# Enhancing Antibiotics Efficacy by Combination of Kuraridin and Epicatechin Gallate with Antimicrobials against Methicillin-Resistant *Staphylococcus aureus*

**DOI:** 10.3390/antibiotics12010117

**Published:** 2023-01-08

**Authors:** Ben Chung-Lap Chan, Nilakshi Barua, Clara Bik-San Lau, Ping-Chung Leung, Kwok-Pui Fung, Margaret Ip

**Affiliations:** 1Institute of Chinese Medicine, State Key Laboratory of Research on Bioactivities and Clinical Applications of Medicinal Plants, The Chinese University of Hong Kong, Shatin, N.T, Hong Kong, China; 2Department of Microbiology, Faculty of Medicine, Prince of Wales Hospital, The Chinese University of Hong Kong, Shatin, N.T, Hong Kong, China; 3School of Biomedical Sciences, Faculty of Medicine, The Chinese University of Hong Kong, Shatin, N.T, Hong Kong, China

**Keywords:** MRSA, epicatechin gallate, kuraridin, gentamicin and fusidic acid resistance, triple therapy

## Abstract

Background: *Staphylococcus aureus* is an opportunistic pathogen and a major cause of nosocomial and community-acquired infections. The alarming rise in Methicillin-resistant *S. aureus* (MRSA) infection worldwide and the emergence of vancomycin-resistant MRSA strains have created an urgent need to identify new and alternative treatment options. Triple combinations of antimicrobials with different antimicrobial mechanisms may be a good choice to overcome antimicrobial resistance. Methods: In this study, we combine two natural compounds: kuraridin from *Sophora flavescens* and epicatechin gallate (ECG) from *Camellia sinensis* (Green tea), which could provide the best synergy with antibiotics against a selected panel of laboratory MRSA with known resistant mechanisms and clinical community-associated (CA) and hospital-associated (HA) MRSA as well. Results: The combined use of ECG and kuraridin was efficacious in inhibiting the growth of a panel of tested MRSA strains. The antibacterial activities of gentamicin, fusidic acid and vancomycin could be further enhanced by the addition of ECG and kuraridin. In time-kill study, when vancomycin (0.5 μg/mL) was combined with ECG (2 μg/mL) and kuraridin (2 μg/mL), a very strong bactericidal growth inhibition against 3 tested strains ATCC25923, MRSA ST30 and ST239 was observed from 2 to 24 h. ECG and kuraridin both possess anti-inflammatory activities in bacterial toxin-stimulated peripheral blood mononuclear cells by suppressing the production of inflammatory cytokines (IL-1β, IL-6 and TNFα) and are non-cytotoxic. In a murine pneumonia model infected with ATCC25923, MRSA ST30 or ST239, the combined use of ECG and kuraridin with vancomycin could significantly reduce bacterial counts. Conclusions: The present findings reveal the potential of ECG and kuraridin combination as a non-toxic herbal and antibiotics combination for MRSA treatment with antibacterial and anti-inflammatory activities.

## 1. Introduction

*Staphylococcus aureus* is an important human opportunistic pathogen and a major cause of a wide range of human infections, ranging from mild skin irritations to severe life-threatening invasive diseases such as necrotizing pneumonia [1]. Staphylococcal super-antigens such as Staphylococcal enterotoxin B (SEB) stimulate the production of proinflammatory mediators by host macrophages causing severe disease and prolonged host inflammatory response [2]. At present, glycopeptide antibiotics, such as vancomycin, have traditionally been the mainstay of treatment of MRSA but overuse has led to the emergence of vancomycin-resistant strains [3]. Hence, alternative therapeutic strategies to identify new agents and paired with existing antibiotics to restore the efficacy against Methicillin-resistant *Staphylococcus aureus* (MRSA) are urgently needed. Herbal sources from Traditional Chinese Medicines (TCM) elaborate a vast array of natural products, either as pure compounds or as standardized plant extracts, providing unlimited opportunities for new drug leads due to the unmatched availability of chemical diversity. It is also commonly accepted that a significant part of this chemical diversity is related to defense mechanisms including resistance to microbiological attack [4]. In our previous studies [5,6,7,8], using a panel of Chinese herbs with a systematic screening of their growth inhibition with a panel of bacteria strains, we have identified some herbs and their active ingredients exhibited promising anti-MRSA activities. For *Sophora flavescens*, we have isolated a non-cytotoxic chalcone named kuraridin, which possessed significant antibacterial effects against a panel of MRSA strains (MIC around 8 μg/mL) [6]. Epicatechin gallate (ECG) from green tea at 20 μg/mL exhibited a four-fold potentiation of the activity of norfloxacin against a norfloxacin-resistant strain of *S. aureus* overexpressing the NorA multidrug efflux pump [9]. Apart from direct and adjuvant antibacterial activities, most of these active ingredients have been shown to possess anti-inflammatory activities which favor their use in anti-MRSA treatment. However, plant antimicrobials are not used as systemic antibiotics directly or as adjuvant with conventional antibiotics in clinics at present. The main reason for this is that their activities alone are not potent enough in the clinical situations when compared with conventionally used antibiotics [10]. To tackle this problem, combinations for better synergy may reduce the agents’ dose and the adverse reactions. Antibiotic combination therapy has long been used in an attempt to improve clinical outcomes, particularly in patients with infections that are associated with high rates of morbidity and mortality such as persistent bacteremia, necrotizing pneumonia, and other deep-seated sites of infections [11]. However, no systematic studies for combinations of promising active ingredients from natural products have been reported. Combinations of ECG and kuraridin at optimal doses as adjuvant therapy with antibiotics may offer an excellent opportunity to maximize clinical outcomes, particularly in the case of antibacterial resistance, and to broaden the spectrum of antibiotic activity.

## 2. Results

### 2.1. The Combination of ECG and Kuraridin Could Potently Inhibit the Growth of MRSA

ECG and kuraridin were first tested on clinical MRSA strains and the MICs of ECG alone and in combination with kuraridin against as a panel of clinical community-associated (CA) and hospital-associated (HA) and the corresponding FICs values were shown in Table 1. The MICs of ECG and kuraridin were 512 and 4–8 μg/mL, respectively, when used alone. In combination with 2 compounds, the MICs of ECG and kuraridin on various tested clinical strains ranged from 0.25 to 8 μg/mL and 1–4 μg/mL, respectively. The synergy outcome was 10/11 and 8/9 in CA and HA MRSA strains, respectively.

### 2.2. The Combination of ECG and Kuraridin Could Enhance the Efficacy of Gentamicin, Fusidic Acid, and Vancomycin Additively against MRSA

ECG and kuraridin were then tested on a panel of MRSA with known antibiotic-resistance mechanisms and the MICs of ECG alone and in combination with kuraridin against and the corresponding FICs values were shown in Table 2. The combination of ECG and kuraridin could synergistically suppress the growth of all tested laboratory strains. In combination with 2 compounds, the MICs of ECG and kuraridin on various laboratory strains ranged from 0.5 to 32 μg/mL and 2–4 μg/mL, respectively. When ECG and kuraridin were combined with the resistant antibiotics on the corresponding laboratory *S. aureus* strains, synergy outcomes were observed in MRSA APH2 (FICI = 0.28) and ANT4 (FICI = 0.38) which were resistant to gentamicin and fusidic acid, respectively. APH2 is highly resistant to gentamicin (MIC > 512 μg/mL). The combination of ECG (0.25 μg/mL) and Kuraridin (2 μg/mL) with gentamicin (16 μg/mL) could overcome the gentamicin resistance of APH2 (Table 2). The combination of ECG (4 μg/mL) and kuraridin (2 μg/mL) with fusidic acid (8 μg/mL) could overcome the fusidic acid resistance of ANT4 strain resistant to fusidic acid (MIC: 64 μg/mL). On the other hand, additively outcomes were observed in MRSA 1199B, RN4220, and APH3 which were resistant to ciprofloxacin, erythromycin, and kanamycin, respectively.

For 3 clinical strains with gentamicin resistance (W231, W233, and W238), ECG (0.25 μg/mL) and kuraridin (2 μg/mL) combined treatment with gentamicin could reduce the MIC of gentamicin in W231 by 64 folds (from 64 to 1 μg/mL) and in W233 and W238 the MIC of gentamicin could be reduced by 32 folds (512 to 16 μg/mL) (Table 3). Synergy outcomes were observed in W231 and W238, and additive outcome in W233. The combination of kuraridin and ECG were also effective in enhancing fusidic acid against 3 clinical fusidic acid-resistant *S. aureus* strains (SA-82356, SA-73621, and SA- 96591) and reduced the MIC of fusidic acid by 32 folds (32 to 1 μg/mL) (Table 4). Additive outcomes were observed in 3 tested strains.

Vancomycin with ECG and kuraridin were also tested on a panel of clinical MRSA strains with MIC = 1 μg/mL (Table 5). The ECG and kuraridin combined treatment could reduce the MIC of vancomycin from 1 to 0.5 μg/mL in 12 tested clinical MRSA strains. Although most of the outcome were additive, the effective dosage of vancomycin was reduced from the micromolar to nanomolar concentration range by ECG and kuraridin.

In time-kill studies using 2 representative clinical strains ST30, ST239 and the standard *S. aureus* strain ATCC25923 (Figure 1), vancomycin (0.5 μg/mL) alone could not effectively inhibit the growth from 0–24 h against three tested strains. A combination of ECG (2 μg/mL) and Kuraridin (2 μg/mL) could effectively inhibit the growth of 3 tested strains and the growth inhibitions on ATCC25923 (Figure 1a) and ST239 (Figure 1c). When vancomycin was combined with ECG and kuraridin, very strong bactericidal growth inhibitions against 3 tested strains were observed from 2 to 24 h. Taken together, a triple combination of vancomycin, ECG and kuraridin were effective in inhibiting the growth on drug-resistant MRSA in vitro, and the in vivo activities of ECG and kuraridin with or without vancomycin against MRSA were further investigated.

### 2.3. ECG and Kuraridin Are Non-Cytotoxic and Could Dose Dependently Inhibit Inflammatory Cytokines Released from Peptidoglycan-Induced and Staphylococcal-Enterotoxin-B-Induced Human Peripheral Blood Mononuclear Cells (PBMC)

Bacterial infections are usually associated with inflammatory cytokines production from immune cells. Hence, the anti-inflammatory effects of ECG and kuraridin on the production of IL-1β, IL-6 and TNF-α stimulated by *S. aureus* isolated peptidoglycan (PGN) and Staphylococcal enterotoxin B (SEB) on human PBMC were investigated. The concentrations of IL-1β, IL-6 and TNFα produced from PGN-stimulated PBMC were 3350 ± 309.6, 58,956 ± 5441 and 4049 ± 92.2 pg/mL, respectively. Kuraridin could also dose-dependently inhibit the production of inflammatory cytokines IL-1β, IL-6 and TNFα s from SEB-stimulated PBMC (Figure 2a). ECG (32–64 μg/mL) could inhibit IL-1β and IL-6 productions from PGN-stimulated PBMC (Figure 2a) but not TNFα production. The anti-inflammatory activities of ECG were weaker when compared with kuraridin. For SEB-stimulated PBMC, the concentrations of IL-1β, IL-6 and TNFα were 510 ± 15.6, 874.5 ± 79.2 and 2349 ± 258.3 pg/mL, respectively. Kuraridin could also dose dependently inhibit the production of inflammatory cytokines IL-1β, IL-6, and TNFα s from SEB-stimulated PBMC (Figure 2b). ECG (64 μg/mL) could only inhibit IL-1β production from SEB-stimulated PBMC (Figure 2b).

The cytotoxicity of the ECG and kuraridin on human PBMC was determined by sodium 3′-[1-(phenylaminocarbonyl)-3, 4-tetrazolium]-bis (4-methoxy-6- nitro) benzene sulfonic acid hydrate (XTT) assay (Figure 2c). ECG (2–64 μg/mL) and kuraridin (2–16 μg/mL) had no significant inhibitory activities on the growth of human PBMC when compared to drug-free control. Growth inhibition of PBMC by less than 15% was observed when the concentrations of kuraridin (32 and 64 μg/mL) were used alone or combined with ECG (32 and 64 μg/mL).

### 2.4. Mouse Pneumonia Model

For mice infected with ATCC25923, MRSA ST239, and ST30, the results were summarized in Figure 3 and Figure 4. Mice intranasal infected with different *S. aureus* strains showed symptoms of severe illness: lethargy, hunched posture, and ruffled fur. Lungs from infected mice were examined 48 h post inoculation (Figure 4a). Tissue sections from lungs infected with *S. aureus* revealed recruitment of immune cells from bronchioles and perivascular areas into the surrounding lung parenchyma. Significant inflammation, bronchial epithelial damage, tissue necrosis, hemorrhage, and different degrees of lesions were observed. For mice infected with ATCC25923 and treated with vancomycin, significant reduction of the mean bacteria counts to 6.064 ± 0.425 log CFU was observed when compared with the control (8.909 ± 0.101 log CFU) (Figure 3a). A combination of ECG and kuraridin could not significantly enhance the antibacterial activities of vancomycin. Slight reduction of bacteria counts was observed in the group treated with ECG and kuraridin (8.502 ± 0.126). Compared with the survival rate of the control group (90%), improvements were observed in all the treatment groups (Figure 3b). Improvement in pneumonia scores (Figure 4b) were observed in of the treatment groups with vancomycin (alone or combined other tested drugs). The tissue profiles of the vancomycin treated group with fewer lesions, infiltration of leukocytes and erythrocytes in the airspace and the alveolar structure were preserved when compared with the control group (Figure 4). Kuraridin in combination with ECG could slightly reduce the log CFU counts (8.25 ± 0.39 and 8.41 ± 0.33, respectively) when compared with the control group (9.24 ± 0.15) (Figure 3a).

The bacterial counts and survival rates (Figure 3b) of vancomycin alone on ST30 infected mice did not differ significantly when used with a combination of ECG, kuraridin, and vancomycin. The pneumonia scores of the lung histology were summarized in Figure 4b. Tissue sections from lungs infected with ST30 revealed the recruitment of leukocytes, inflammation in the lung parenchyma, bronchial epithelial damage, and tissue necrosis (Figure 4a). Inflammation, bronchial epithelial damage, tissue necrosis, hemorrhage and lesion sizes caused by ST239 infection were severe (average pneumonia score = 4) (Figure 4b). The tissue profiles of the treatment groups (vancomycin alone, ECG + vancomycin, Kuraridin + vancomycin, and ECG + Kuraridin + vancomycin) were with fewer lesions, infiltration of leukocytes and erythrocytes in the airspace and the alveolar structure were preserved when compared with the control group.

For ST239-infected mice treated with vancomycin, a significant reduction of the mean bacteria counts to 4.73 ± 0.26 log CFU was observed when compared with the control (7.39 ± 0.27 log CFU) (Figure 3a). Combinations of ECG and kuraridin could not enhance the antibacterial activities of vancomycin. A slight reduction of bacteria counts was observed in the group treated with ECG and kuraridin (6.40 ± 0.30). The survival rates of all groups are similar and 1 mouse died in the control group, the groups treated with ECG + kuraridin (Figure 3b). Tissue sections from lungs infected with ST239 revealed mild recruitment of immune cells from bronchioles and perivascular areas into the surrounding lung parenchyma when compared with ATCC25923 and ST30-infected mice (Figure 4a). Inflammation, bronchial epithelial damage, tissue necrosis, hemorrhage and lesion sizes caused by ST239 infection were also less severe (average pneumonia score = 2) when compared with mice infected with ATCC25923 and ST30 (average pneumonia score = 4). Improvements in pneumonia score (Figure 4b) were observed in the treatment groups with vancomycin (alone or combined with other tested drugs). The tissue profiles of those groups with fewer lesions, infiltration of leukocytes and erythrocytes in the airspace and the alveolar structure were preserved when compared with the control group (Figure 4a).

## 3. Discussion

Our study found that combined use of ECG and kuraridin was efficacious in inhibiting the growth of a panel of tested MRSA strains in vitro. ECG and kuraridin are non-cytotoxic and possess anti-inflammatory activities. By using specific strains with known antibiotic resistant mechanisms, the antibacterial activities of gentamicin, fusidic acid, and vancomycin could be further enhanced by the addition of ECG and Kuraridin in vitro. Time-kill studies showed that the antibacterial activities of vancomycin with ECG and Kuraridin were bactericidal, and the combination was better than vancomycin or ECG/Kuraridin when used separately. The dosage of vancomycin could be reduced to therapeutically relevant concentrations (nanograms level). Vancomycin was associated with many side effects, including vestibular and renal toxicity [12]. Apart from their direct use, kuraridin and ECG may be a good choice to supplement with the sub-optimal dosage of vancomycin to prevent its side effects and drug resistance in MRSA treatment. It is a pity that ECG and kuraridin could not enhance vancomycin in animal testing. The reason may be due to the fact that the concentrations of ECG and kuraridin were not high enough to reach the site of infection. In this regard, the mice pneumonia model may not be an ideal model for evaluating the antibacterial effects of the current study. Some other animal models such as skin and wound infection models may be useful in evaluating the in vivo efficacy of ECG and kuraridin. We hope to confirm the antibacterial activities of ECG and kuraridin using these models with MRSA in further study.

Another reason that ECG and kuraridin could not enhance vancomycin in animal testing, which may be due to low bioavailability of ECG and Kuraridin. Green tea extract alone or in combination with amoxicillin has been shown to weaken the antibacterial effect of amoxicillin in MRSA-infected mice, and tea drinking is not recommended in combination with amoxicillin treatment [13]. In that study, the mice were intraperitoneally infected with MRSA and amoxicillin and green tea extract were administered via gastric perfusion. They found that MICs of amoxicillin were greatly decreased in the presence of 0.25% tea extract. Apart from animal testing, green tea catechin and soy isoflavones administrations have also been shown to reduce the bioavailability of statins in human [14]. In two open-label, single-dose, three-phase clinical pharmacokinetic studies, healthy Chinese male subjects were given a single dose of rosuvastatin 10 mg (Study A) or simvastatin 20 mg (Study B) on 3 occasions: 1. without herbs; 2. with green tea extract; 3. with soy isoflavone extract. In study A (n = 20), intake of green tea extract significantly reduced the systemic exposure to rosuvastatin by nearly one-third. In study B (n = 18), intake of soy isoflavones was associated with reduced systemic exposure to simvastatin acid. Taken together, these studies suggested that repeated green tea catechin or soy isoflavonones administration could reduce the bioavailability of statins in healthy volunteers and these effects might be predicted to reduce the beneficial action of the drugs. In our current study, ECG or kuraridin at the tested concentrations did not significantly enhance or decrease the efficacy of vancomycin in the current study. ECG and kuraridin are natural flavonoids that may affect the bioavailability of vancomycin and hence the expected synergistic antibacterial activities were not observed.

Triple combination of some other natural products with antibiotics have been shown to be effective against drug-resistant bacteria. A combination of propyl gallate (25 μg/mL) and octyl gallate (6.25 μg/mL) has been shown to improve the MIC of oxacillin against a panel of MRSA strains from 64–256 μg/mL to 2 μg/mL [15]. Palmitic acid (0.3–1.3 mg/mL) with surfactant Span85 (0.08–0.4%) with oxacillin (15–100 μg/mL) has been shown to inhibit the growth of a panel clinical MRSA strains in vitro [16]. ECG has been shown to possess a high affinity for the positively charged Staphylococcal membrane and induced changes to the biophysical properties of the bilayer that are likely to account for its capacity to disperse the cell wall biosynthetic machinery responsible for β-lactam resistance [17]. For kuraridin, the inhibitory mechanism against MRSA was not yet known, but different studies showed that kuraridin could inhibit a wide array of enzymes activities, namely protein tyrosine phosphatase 1B [18], beta-site APP cleaving enzyme 1 (BACE1 and cholinesterases [19], aldose reductase, tyrosinase and melanin synthesis [20], glycosidase [21], diacylglycerol acyltransferase [22], and tyrosinase [23]. The antibacterial activities of kuraridin may involve inhibiting some key enzymes in MRSA for its survival. Further studies are required to investigate this issue. Ser/Thr phosphorylation/dephosphorylation is a common theme in the regulation of cellular functions determining metabolic activity and virulence also in the major human pathogen *S. aureus* [24]. One of the target enzymes which may be inhibited by kuraridin is sortase A (SrtA). *S. aureus* uses the SrtA to display surface virulence factors suggesting that compounds that inhibit its activity will function as potent anti-infective agents. In *S. aureus*, more than 20 distinct surface virulence factors are anchored to the cell wall by the extracellular SrtA [25]. In our coming studies, the effects of kuraridin on SrtA will be investigated.

## 4. Materials and Methods

### 4.1. Antibiotics, Epicatechin Gallate (ECG) and Kuraridin

Ciprofloxacin, erythromycin, fusidic acid, gentamicin, kanamycin, and vancomycin, the major classes of antibiotics typically used in MRSA treatment, were used. ECG (CAS number: 1257-08-5) and kuraridin (CAS number: 34981-25-4) were purchased from SR Pharmasolutions (Hong Kong, China). The structures of the compounds were confirmed by mass spectrometry; the purity of ECG and kuraridin were >95% confirmed by HPLC. The MIC of kuraridin was similar to our previously published results on Sophora flavescens isolated Kuraridin [6]. All other chemicals were purchased from Sigma Chemical Company (St. Louis, MO, USA).

### 4.2. Bacterial Strains and Preparation of Bacteria Culture

A total of 29 *S. aureus* strains were used in the study. The MRSA strains with different antibiotic resistance and with known resistance mechanisms were used (Table 6). A methicillin-sensitive strain: *S. aureus* ATCC25923 was used as a control strain. Three clinical strains, SA-82356, SA-73621, and SA-96591 resistant to fusidic acid, were included to determine the enhancement effect of the natural products on fusidic acid. Twenty non-duplicate hospital-associated (HA)-MRSA (MRSA W231-W240) and community-associated (CA)-MRSA clinical isolates (MRSA W44-W48, W101, W103, W106, W113, W114, and ST30) were also used for screening. ST239, a representative strain of healthcare-associated multidrug-resistant and ST30 a representative strain of community-associated multidrug-resistant MRSA strain prevalent in Asian countries were included in the study.

### 4.3. Checkerboard and Time Kill Assay

A checkerboard assay [11] was conducted to measure the synergy for ECG and kuraridin (2 compounds) against community-associated (CA) and hospital-associated (HA) MRSA strains, serial 2-fold dilutions of ECG and kuraridin were mixed in each well of a 96-well microtiter plate so that each row (and column) contained a fixed amount of one agent and increasing amounts of the second agent. Stock solutions of the tested compounds were made in Müller-Hinton (MH) broth. The resulting plate presents a pattern in which every well contains a unique combination of concentrations between the two molecules. The concentrations of kuraridin ranged from 0 to 64 μg/mL, while ECG concentrations ranged from 0 to 512 μg/mL. Each microtiter well was inoculated with approximately 10^5^ CFU/mL of bacteria, and the plates were incubated at 37 °C for 24 h under aerobic conditions. MIC values obtained for a given combination were used to evaluate the effects of the combination between ECG and kuraridin by calculating the fractional inhibitory concentration index (FICI) using fractional inhibitory concentration index (FIC) and the following formula:FIC of ECG = MIC ECG in combination/MIC of ECG alone; FIC of kuraridin = MIC of kuraridin in combination/MIC of kuraridin alone; hence FICI = FIC of ECG + FIC of kuraridin.

Off-scale MICs were converted to the next highest or next lowest doubling concentration. The lowest FICI calculated value obtained for a given strain was reported in Table 1, Table 2, Table 3, Table 4 and Table 5, with the corresponding MICs values. “Synergy” was defined when FIC index was less than or equal to 0.5; while “additive” in which the FIC index was greater than 0.5 and less than or equal to 1.0; whereas “indifferent” when the FIC index was greater than 1.0 and less than or equal to 2.0; and “antagonistic” in cases which the FIC index was greater than 2.0 [17].

Triple combination checkerboard assay [15]of the ECG and kuraridin with the antibiotics (ciprofloxacin, erythromycin, fusidic acid, gentamicin, kanamycin, and vancomycin) against gentamicin-resistant MRSA strains, fusidic acid-resistant MRSA strains and clinical MRSA strains was performed to identify the best combinations of the chosen natural products with the antibiotics. Fractional inhibitory concentration (FIC) indices for triple combinations [15] were calculated as the following equation:FIC of antibiotics = MIC antibiotics in combination/MIC of antibiotics alone; FIC of ECG = MIC ECG in combination/MIC of ECG alone; FIC of kuraridin = MIC of kuraridin in combination/MIC of kuraridin alone; hence FICI = FIC of antibiotics + FIC of ECG + FIC of kuraridin

Off-scale MICs were converted to the next highest or next lowest doubling concentration. The lowest FICI calculated value obtained for a given strain was reported in Table 2, Table 3, Table 4 and Table 5 with the corresponding MICs values. The concentrations of antibiotics, ECG, and kuraridin were ranged from 0 to the MIC (μg/mL).

For time-kill curves, the bacteria were grown with the natural compounds and antibiotics combinations; and normal saline (as control) with bacteria in the Müller-Hinton (MH) medium were grown for 24 h [17]. An overnight broth culture was diluted to obtain a starting inoculum of about 1 × 10^5^ CFU/mL. Vancomycin, ECG and kuraridin alone or in combination were added to the bacterial broth and incubated at 37 °C. The time-kill curves for the bacteria were determined by CFU counting. Ten microliters of broth from each preparation were taken at 0 h and after 2, 4, 8, and 24 h of incubation for bacterial counts. Each aliquot was serially diluted and plated onto MH agar plates and incubated overnight at 37 °C for 18–24 h, and the number of CFU/mL was determined. The results were collected from 3 independent experiments.

### 4.4. Anti-Inflammation Effects and Cytotoxicity of Epicatechin Gallate (ECG) and Kuraridin

The anti-inflammatory actions of ECG and kuraridin were evaluated using human peripheral blood mononuclear cells (PBMCs) and human cultured macrophages. PBMCs were isolated from the buffy coat of healthy adult donors (Red Cross, Hong Kong SAR, China) by Ficoll-Paque Plus density gradient (Amersham Biosciences, Uppsala, Sweden) following the manufacturer’s instruction. PBMCs were adjusted to a concentration of 2 × 10^6^ cells/mL in a falcon tube, and 100 μL aliquots of cell suspension were placed in a culture plate. PBMCs were cultured without stimulation or incubated with kuraridin or ECG (0, 4, 8, 14, 32 and 64 μg/mL) for 30 min, Staphylococcal endotoxin B (SEB) or peptidoglycan (PGN) were then added to stimulate the cells for further 24 h in a 95% humidified air incubator containing 5% CO_2_ at 37 °C. Dexamethasone at 1 μg/mL was used as a positive control to inhibit the production of inflammatory cytokines. The supernatants were collected, and the concentration of inflammatory and T cells associated cytokines, including TNF-α, IL-1β and IL-6 were determined by human cytokine ELISA kits (B.D. Biosciences, San Diego, CA, USA) following the manufacturer’s instruction with detection limits that ranged from 3.1 to 7.8 pg/mL. The percentage inhibition of cytokine was calculated by the following formula:Percentage (%) inhibition = (1 − (Concentration of cytokine with drug and stimulant/Concentration of cytokine with stimulant only)) × 100%

The cytotoxicity of ECG and kuraridin was determined by (Sodium 3′-[1-(phenylaminocarbonyl)-3, 4-tetrazolium]-bis (4-methoxy-6-nitro) benzene sulfonic acid hydrate XTT assay using buffy coat purified human peripheral mononuclear cells (PBMCs) collected from Hong Kong Red Cross. Cells were plated in 96-well plates at a density of 10^5^ cells/well. Serial dilutions of the compounds were added to the wells. The plates were maintained in a humidified incubator at 37 °C and 5% CO_2_. After 3 days, 50 μL of XTT/Phenazine methosulfate (PMS) solution (20 μM) was added to each well. Then, the plates were incubated at 37 °C for 4 h. The absorbance of the wells was determined by a spectrophotometer (DTX880 Multimode detector) (Brea, CA, United States) at 450 nm. The toxicity represents the ratio of absorbance of a well in the presence of compounds with the absorbance of control wells in the presence of a medium containing 1% DMSO.

### 4.5. Mouse Pneumonia Model

Murine lung infection model [26] was used to validate the in vivo efficacy of ECG and kuraridin in combination with vancomycin against MRSA. The animal study protocols were approved by the Animal Experimentation Ethics Committee of the Chinese University of Hong Kong (Ref. no. 12/071/MIS). Balb/c mice aged 7–9 weeks were anesthetized and inoculated with 1 × 10^7^ to 2 × 10^8^ selected MRSA and standard strains: Panton-Valentine leukocidin positive CA-MRSA strain (ST30) and ATCC25923 in a volume of 20 μL intranasally. The freshly prepared stock solution of kuraridin was dissolved in ethanol and further diluted with saline. The final concentration of kuraridin (12 mg/mL) contained 20% ethanol. ECG and vancomycin were dissolved in saline and the final concentrations were 12 mg/mL and 12 mg/mL, respectively. A vehicle control group, which was infected with MRSA and received diluted ethanol in saline, was included.

One hundred and twenty-five mice were included in the study. Five mice were used as normal control without infection and treatment. One hundred and twenty mice were equally divided into three groups (n = 40 for each group) and infected with ATCC25923, ST30 or ST239. Each group of mice was subdivided into four treatment options (n = 10 for each subgroup): no treatment, vancomycin (60 mg/kg) only, ECG (120 mg/kg) and kuraridin (120 mg/kg) and kuraridin (120 mg/kg), vancomycin (60 mg/kg) and ECG (120 mg/kg). Treatment and control regimens were initiated 2 h post-inoculation every 12 h two times daily for 2 days. All antimicrobials (0.1 mL) were administered subcutaneously [16,17]. Mortality of control and therapeutic groups were recorded during 48 h of therapy. The animals were then sacrificed by cervical dislocation. The left lung of the animals used for bacteriological analyses was homogenized in saline (0.1 g of tissue to a final volume of 1 mL), serially diluted, and cultured on blood agar plates. In the next phase, the best combinations of natural products/antibiotics at the sub-MIC dosage and single MIC dosage of the corresponding MRSA-sensitive antibiotics as positive control were chosen with reference to the initial phase results for the synergistic studies using this animal model. ST30, and ATCC25923 were used. The ATCC25923 strain did not effectively infect in BALB/c mice; even the inoculations were at 1 × 10^9^ CFU. For infecting this bacterial strain in BALB/c mice, neutropenia was induced in the mice prior to the infection using cyclophosphamide (intraperitoneally injection with 150 and 100 mg/kg cyclophosphamide at 96 and 24 h before infection, respectively) [27]. After treatment, the animals (n = 10 for each group) were then sacrificed by cervical dislocation; the lungs were dissected and removed under aseptic conditions for bacterial loading and pneumonia assessment. Left lungs were used for bacteriological, and cytokine analyses were homogenized in saline (0.1 g of tissue to a final volume of 1 mL), serially diluted, and cultured on blood agar plates. The right lung was used for pneumonia assessment by histology.

For pneumonia assessment [26], the right lungs of the animals were perfused with 1 mL of 10% neutral-buffered formalin. The tissues were dehydrated and embedded in paraffin, cut, and stained with hematoxylin and eosin. The stained sections were examined by light microscopy to assess the level of inflammation. The evidence of pneumonia was determined by histopathological and was scored (0–5) according to the levels of leukocyte and erythrocyte infiltration, alveolar integrity and epidermis damage: (Score = 0)—No lesions, no leukocyte or erythrocyte infiltrate & normal epithelia; (Score = 1)—No lesions, except some leukocyte infiltration; (Score = 2)—No lesions, some leukocytes and erythrocytes in the airspace but the alveolar structure is preserved; (Score = 3)—1 to 2 lesions smaller than 500 μm in length/width or many smaller ones, some leukocyte and erythrocyte infiltrate in the alveoli, but the alveolar structure is preserved, no epithelial damage; (Score = 4)—Less than 3 lesions smaller than 1000 μm in length/width; leukocytes and erythrocytes throughout the lesion, alveolar structure not preserved within the lesion, some epithelial damage; (Score = 5)—More than 3 lesions 1000 μm in length/width, leukocytes and erythrocytes throughout the lesion, alveolar structure not preserved within the lesion.

### 4.6. Statistical Analysis

Statistical analyses and significance, as measured by the student’s *t*-test were performed using GraphPad PRISM software version 8.0 (GraphPad Software, San Diego, CA, USA). For histopathological analyses, Mann–Whitney test was used. In all comparisons, *p* < 0.05 was considered statistically significant.

## 5. Conclusions

In conclusion, the present study reported for the first time that the combined use of ECG and kuraridin could enhance the anti-bacterial activities of gentamicin, fusidic acid, and vancomycin against MRSA in vitro. In mice pneumonia infection model, the combined use of ECG and kuraridin could slightly reduce bacterial counts in MRSA-infected mice. With the success in MRSA treatment, kuraridin, and ECG combined treatment may be useful in other drug resistance bacterial treatment.

## 6. Patents

The work reported in this manuscript was filed in the United States Patent (App. 16/684,073 number: 202111143384.9).

## Figures and Tables

**Figure 1 antibiotics-12-00117-f001:**
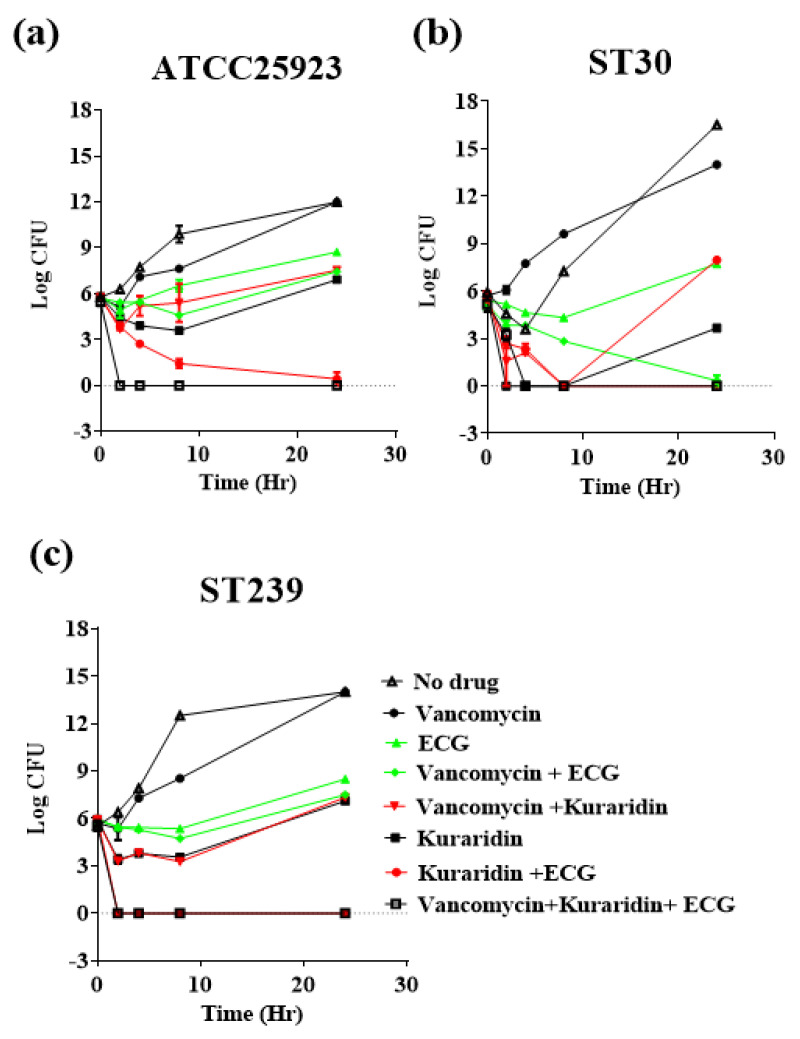
In vitro time-kill curves (**a**) ATCC25923, (**b**) ST30 and (**c**) ST239; epicatechin gallate (ECG) (2 μg/mL), kuraridin (2 μg/mL), and vancomycin (0.5 μg/mL) alone and in combination. Results are expressed as log CFU/mL and are given as mean ± standard error of mean (n = 3).

**Figure 2 antibiotics-12-00117-f002:**
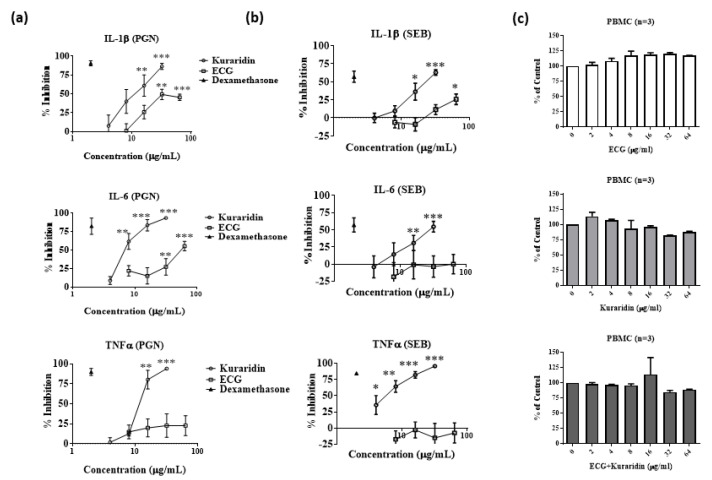
Inhibitory effects of epicatechin gallate (ECG) and kuraridin on inflammatory cytokines (IL-1β, IL-6 and TNFα) production and cytotoxicity of human peripheral blood mononuclear cells (PBMC) stimulated with (**a**) Peptidoglycan (PGN) and (**b**) Staphylococcal enterotoxin B (SEB) (n = 4). Dexamethasone at 1 μg/mL was used as positive control to inhibit the production of inflammatory cytokines. PGN (0.1 μg/mL) or SEB (10 μg/mL) was used to stimulate the PBMC to produce cytokines. Kuraridin or ECG (4–64 μg/mL) was added to the cells and the cell supernatant was collected for cytokine assays. (**c**) Cellular toxicity (XTT assay) of epicatechin gallate (ECG) and kuraridin on human peripheral blood mononuclear cells (PBMC) (n = 3). Significant results by comparing the drug treatment groups with the drug-free control are indicated (* *p* < 0.05; ** *p* < 0.01; *** *p* < 0.001).

**Figure 3 antibiotics-12-00117-f003:**
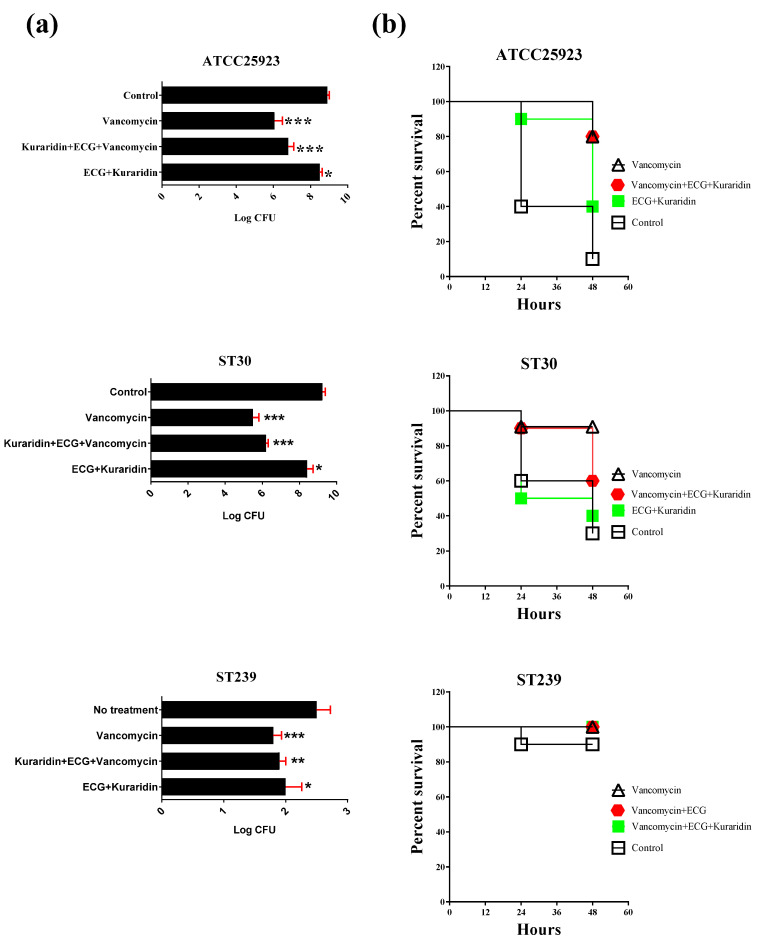
Mice infected with ATCC25923, ST239, or ST30 (3 × 10^8^ colony forming unit (CFU)) with different treatment options. Mice were sacrificed after 48 h. (**a**) The bacteria counts in Log CFU recovered from the left lungs; (**b**) the survival rates of each treatment group. Significant results by comparing the drug treatment groups with the drug free control are indicated (* *p* < 0.05; ** *p* < 0.01; *** *p* < 0.001) (n = 10).

**Figure 4 antibiotics-12-00117-f004:**
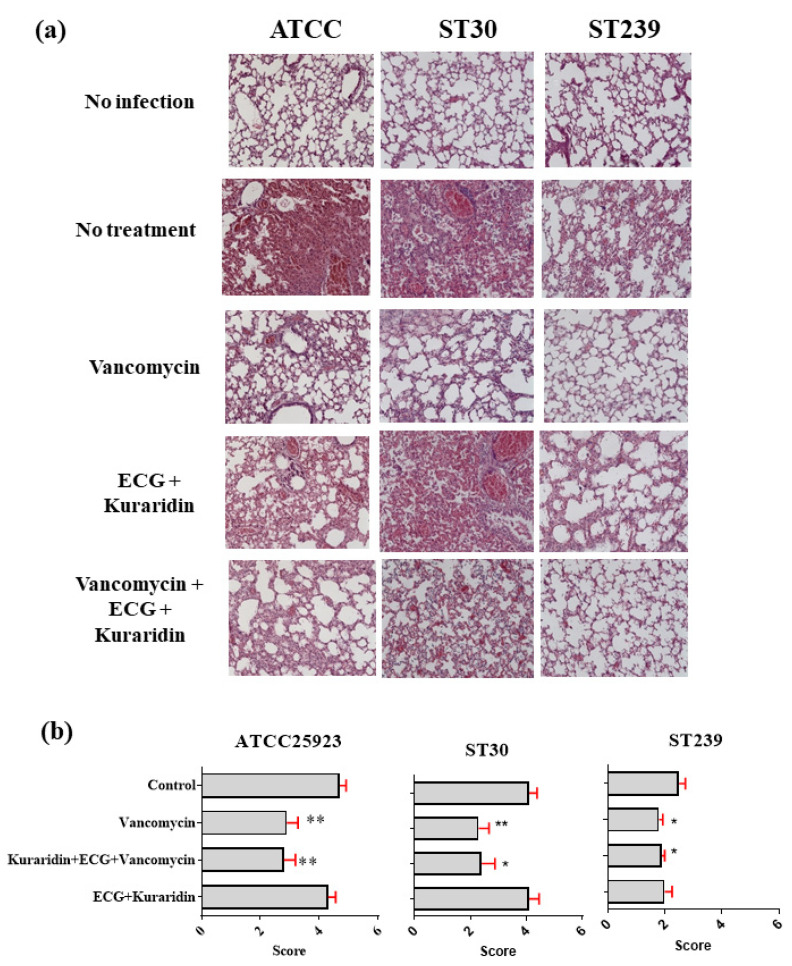
Mice infected with ATCC25923, ST239, or ST30 (3 × 10^8^ colony forming unit (CFU)) with different treatment options. Mice were sacrificed after 48 h. (**a**) Histology of lung tissue in different groups; (**b**) the pneumonia scores of the right lung histology (n = 10). Significant results by comparing the drug treatment groups with the drug free control are indicated (* *p* < 0.05; ** *p* < 0.01 (n = 10).

**Table 1 antibiotics-12-00117-t001:** Minimum inhibitory concentrations (MIC) (μg/mL) and fractional inhibitory concentration indices (FICI) of kuraridin (Kur) and epicatechin gallate (ECG) against community-associated (CA) and hospital-associated (HA) MRSA strains.

Strains	Category	MIC Alone (μg/mL)		MIC Combine (μg/mL)		FIC		FICI	Outcome
		ECG	Kur	ECG	Kur	ECG	Kur		
W44	CA MRSA	512	8	2	2	0.004	0.25	0.25	Synergy
W45	CA MRSA	512	8	1	4	0.002	0.5	0.50	Synergy
W46	CA MRSA	512	8	8	4	0.015	0.5	0.52	Additive
W47	CA MRSA	512	8	2	4	0.004	0.5	0.50	Synergy
W48	CA MRSA	512	8	1	4	0.002	0.5	0.50	Synergy
W101	CA MRSA	512	8	1	4	0.002	0.5	0.50	Synergy
W103	CA MRSA	512	8	1	4	0.002	0.5	0.50	Synergy
W106	CA MRSA	512	8	1	2	0.002	0.25	0.25	Synergy
W113	CA MRSA	512	8	1	4	0.002	0.5	0.50	Synergy
W114	CA MRSA	512	8	1	4	0.002	0.5	0.50	Synergy
ST30	CA MRSA	512	8	0.25	4	0.0004	0.5	0.50	Synergy
W231	HA MRSA	512	4	8	1	0.016	0.25	0.27	Synergy
W232	HA MRSA	512	8	4	2	0.008	0.25	0.26	Synergy
W233	HA MRSA	512	4	2	2	0.004	0.5	0.50	Synergy
W234	HA MRSA	512	8	1	4	0.002	0.5	0.50	Synergy
W235	HA MRSA	512	8	1	4	0.002	0.5	0.50	Synergy
W238	HA MRSA	512	8	1	4	0.002	0.5	0.50	Synergy
W239	HA MRSA	512	8	1	4	0.002	0.5	0.50	Synergy
W240	HA MRSA	512	8	4	4	0.008	0.5	0.51	Additive
ST239	HA MRSA	512	8	2	4	0.004	0.5	0.50	Synergy
ATCC 25923	MSSA	512	8	4	4	0.008	0.5	0.51	Additive

**Table 2 antibiotics-12-00117-t002:** Minimum inhibitory concentrations (MIC) (μg/mL) and fractional inhibitory concentration indices (FICI) of kuraridin (Kur) and epicatechin gallate (ECG) with or without antibiotics against the laboratory *S. aureus* strains with known antibiotic-resistant mechanisms.

	MIC Alone (μg/mL)			MIC Combine (μg/mL)			FIC			FICI	Outcome
Strains/ Antibiotics (An)	ECG	Kur	An	ECG	Kur	An	ECG	Kur	An		
1199B	512	16	16	32	4	8	0.06	0.25	0.5	0.81	Additive
Ciprofloxacin	512	16	-	32	4	-	0.06	0.25	-	0.31	Synergy
RN4220	512	8	256	8	2	128	0.01	0.26	0.5	0.76	Additive
Erythromycin	512	8	-	8	2	-	0.01	0.26		0.5	Synergy
APH2	512	8	512	0.25	2	16	0.0005	0.25	0.0312	0.28	Synergy
Gentamicin	512	8	-	0.5	2	-	0.0009	0.25	-	0.25	Synergy
APH3	512	8	256	0.5	2	128	0.0009	0.25	0.5	0.75	Additive
Kanamycin	512	8	-	0.5	2	-	0.0009	0.25		0.25	Synergy
ANT4	512	8	64	4	2	8	0.0007	0.25	0.125	0.38	Synergy
Fusidic acid	512	8	-	16	2		0.03	0.25		0.28	Synergy

**Table 3 antibiotics-12-00117-t003:** Minimum inhibitory concentrations (MIC) (μg/mL) and fractional inhibitory concentration indices (FICI) of kuraridin and epicatechin gallate (ECG) with or without gentamicin against gentamicin (Gen) resistant MRSA strains.

Strains	MIC Alone (μg/mL)			MIC Combine (μg/mL)			FIC			FICI	Outcome
	ECG	Kur	Gen	ECG	Kur	Gen	ECG	Kur	Gen		
W231	512	8	64	0.25	2	1	0.0004	0.25	0.015	0.27	Synergy
	512	8	-	4	2	-	0.008	0.25	-	0.26	Synergy
W233	512	8	512	0.25	4	16	0.0005	0.5	0.03	0.53	Additive
	512	8	-	16	8	-	0.03	1	-	1.03	Indiffferent
W238	512	8	512	0.25	2	16	0.0004	0.25	0.03	0.28	Synergy
	512	8	-	0.5	4	-	0.0004	0.5	-	0.50	Synergy

**Table 4 antibiotics-12-00117-t004:** Minimum inhibitory concentrations (MIC) (μg/mL) and fractional inhibitory concentration indices (FICI) of kuraridin and epicatechin gallate (ECG) with antibiotics against fusidic acid (Fus) resistant MRSA strains.

Strains	MIC Alone (μg/mL)			MIC Combine (μg/mL)			FIC			FICI	Outcome
	ECG	Kur	Fus	ECG	Kur	Fus	ECG	Kur	Fus		
82356	512	8	32	0.25	4	1	0.008	0.25	0.03	0.53	Additive
	512	8	-	8	4	-	0.016	0.5	-	0.52	Additive
73621	512	8	32	0.25	4	1	0.004	0.5	0.03	0.53	Additive
	512	8	-	16	4	-	0.003	0.5	-	0.53	Additive
96591	512	4	32	0.25	2	1	0.002	0.5	0.03	0.53	Additive
	512	8	-	16	4	-	0.003	0.5	-	0.53	Additive

**Table 5 antibiotics-12-00117-t005:** Minimum inhibitory concentrations (MIC) (μg/mL) and fractional inhibitory concentration indices (FICI) of kuraridin and epicatechin gallate (ECG) with antibiotics against clinical MRSA strains.

Strains	MIC Alone			MIC Combine			FIC			FICI	Outcome
	ECG	Kur	Van	ECG	Kur	Van	ECG	Kur	Van		
W44	512	8	1	0.25	2	0.5	0.0004	0.25	0.5	0.75	Additive
W45	512	8	1	0.25	2	0.5	0.0004	0.25	0.5	0.75	Additive
W46	512	8	1	1	4	0.5	0.002	0.5	0.5	1.00	Additive
W47	512	8	1	0.25	2	0.25	0.0004	0.25	0.25	0.50	Synergy
W48	512	8	1	1	2	0.5	0.002	0.25	0.5	0.75	Additive
W231	512	8	1	0.25	1	0.5	0.0004	0.125	05	0.62	Additive
W232	512	8	1	8	2	0.5	0.03	0.25	0.5	0.75	Additive
W233	512	8	1	8	2	0.5	0.03	0.25	0.5	0.75	Additive
W238	512	8	1	0.25	2	0.5	0.0004	0.25	0.5	0.75	Additive
W239	512	8	1	0.5	2	0.5	0.0009	0.25	0.5	0.75	Additive
ST30	512	8	1	0.25	1	0.5	0.0004	0.125	0.5	0.62	Additive
ST239	512	8	1	0.25	1	0.5	0.0004	0.125	0.5	0.62	Additive

**Table 6 antibiotics-12-00117-t006:** The *S. aureus* strains with known resistance mechanism used in the study.

*S. aureus* Strains	Resistance	Mechanism
1199B	ciprofloxacin	overexpression of the NorA efflux pump
RN4220-pUL5054	erythromycin	multicopy plasmid pUL5054 coding for MsrA
APH2″-AAC6′	gentamicin	experimentally induced aminoglycosides resistance by methylation of specific nucleotides within the A-site of rRNA (aminoglycoside- 6′-N-acetyltransferase/2″-O- phosphoryl transferase)
APH3′	kanamycin	experimentally induced aminoglycosides resistance by methylation of specific nucleotides within the A-site of rRNA (aminoglycoside-3′-O- phosphoryl transferase)
ANT4′	fusidic acid	experimentally induced aminoglycosides resistance by methylation of specific nucleotides within the A-site of rRNA (aminoglycoside-4′-O- phosphoryl transferase) is
MSSA ATCC25923 ^a^	methicillin-sensitive	Control strain

^a^ Methicillin-sensitive strain (MSSA) used as a control for the current study.

## Data Availability

The data presented in this study are available on request from the corresponding author.

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
