# Peer review of "Enhancing Antibiotics Efficacy by Combination of Kuraridin and Epicatechin Gallate with Antimicrobials against Methicillin-Resistant Staphylococcus aureus"

_antibiotics, 2023, doi:10.3390/antibiotics12010117_

Round 1
Reviewer 1 Report
The manuscript is interesting but each of the suggestions below should be considered so that it can be a potential candidate for publication.
Introduction
L.78-81. This paragraph seems to be a result. An introduction is supposed to end with the objective of the investigation. It is better that the paragraph be removed from this section.
Results
L.87-88. This applies to the combination, please put it as the MIC of ECG and kuraridin by themselves is much higher.
L.91-92. In fact, for CA it was 10/11, not 9/10. For HA is correct 9/10
L.97-107. The description in this paragraph is found in the materials and methods section. Please delete it.
L.100. calculating.
L.109. Please correct the number of bacterial strains that were used since 29 are included in the methodology and 32 when described in the paragraph.
Table 1. Include the units (mcg/mL). What were the tested concentrations of ECG and KUR, as well as the combination? Could you explain in more detail how the determination of MIC combined is performed? Where did the authors get the number 32 and 4 in the first case (1199B strain)?
L.134. Delete "against and" from "with gentamicin against and reduced the"
Table 2. Why are the results only found in 20 strains? and the others?
L.156. delete "against three tested strains" or make sense to the sentence since it is repetitive.
L.157-158. The sentence is not understood. Please, it is important to rewrite it.
L.159. It is repetitive "with vancomycin (0.5 g/mL)" since it was mentioned before.
L.168. against what? complete the sentence or make sense.
L.182. Delete "and"
L.192 (Figure 3). Positive or negative? if positive, cytokine concentrations would be elevated in this group. Please check it out.
L.216. Why were antibiotics such as gentamicin or fusidic acid not tried in this animal model?
L.241-242. This sentence is repetitive.
L.250-251. This sentence is repetitive.
Discussion
The discussion is very deficient considering the results presented. Therefore, it has to be completely restructured considering the results obtained and having a much more scientific basis.
L.262. Delete "additively".
L.265. Change "alone" to "separately"
L.271. low bioavailability of ECG and Kuraridin
L.274. Change "In this study" to "In that study"
Materials and methods
L.308-309. Why is it mentioned that Kanamycin, erythromycin and ciprofloxacin were tested as antibiotics and there are no results in the article? Who provided the antibiotics for this study? Include the reference with catalog number.
L.313. Change "The MIC of kuraridin is similar " to "The MIC of kuraridin was similar "
L.314. Include the reference.
L.318. Please, be more explicit in the number of bacterial strains that were used since those found in the paragraph add up to 32 strains.
L.352. Where did the authors get this criterion (FICI<0.5)? Please reference?.
L.360. Change "normal saline" to "saline".
L.360. Include the meaning of MH, please.
L.364. Why at 35 ºC if the optimum temperature for S. aureus is 37 ºC?
L.374-376. It is important that the methodology be described in more detail, for example, the treatments (experimental groups, how many and which are they?) and the concentrations or doses at which the treatments were tested.
L.387. The plates were maintained in an incubator at 37 C and 5% CO2.
L.393. cell viability of at least 85% considered the compounds non-toxic. Why 85% and not 90% or 80%? What did the authors base their definition of 85% on?
L.399-402. How many mice were used in the experiment? How many experimental groups did the experiment consist of?
L.402-404. It is important that the way in which the treatments were prepared is described. For example, how was the Kuraridin solution prepared? Was only ethanol used? What are the concentrations of the solutions? Why was only vancomycin tested as an antibiotic (chemical entity)?
L.408-411. Was it only 3 experimental groups? And the combinations to evaluate synergism? What were the combinations?
L.422. Change "on" to "at".
L.422-423. Could the authors explain why the waiting times of 96h and 24h before carrying out the infection?
Statistical analysis
L.447. Histopathological analyzes cannot be analyzed with parametric statistics since the variables are qualitative (categorical). Please carry out the corresponding analysis (NON-parametric statistics).
Conclusions
L.455-458. conclusion is not entirely true since in reality the combination of vancomycin with ECG and kuraridin did not present significant (synergistic) effects. Contradicts what is mentioned in L.216

Author Response
We appreciate the reviewer's comments and constructive feedback related to the manuscript. We revised the manuscript according to these comments and provide detailed answers to the comments below. All changes in the manuscript can be tracked via the track change function.
Introduction
L.78-81. This paragraph seems to be a result. An introduction is supposed to end with the objective of the investigation. It is better that the paragraph be removed from this section.
We have removed that paragraph from the introduction.
Results
L.87-88. This applies to the combination, please put it as the MIC of ECG and kuraridin by themselves is much higher.
We have revised the paragraph and revised Table 1 to Tables 1 & 2 and added the MIC values of both alone and combine.
L.91-92. In fact, for CA it was 10/11, not 9/10. For HA is correct 9/10
We have revised the values, the synergy outcome should be 10/11 and 8/9 in CA and HA MRSA strains respectively.
L.97-107. The description in this paragraph is found in the materials and methods section. Please delete it.
L.100. calculating.
We have deleted the corresponding paragraph.
L.109. Please correct the number of bacterial strains that were used since 29 are included in the methodology and 32 when described in the paragraph.
We have only included 29 strains in the study. W231, W233, and W238 are gentamicin-resistant strains, which were from our selected HA MRSA panel and appeared in two MIC tests.
Table 1. Include the units (mcg/mL). What were the tested concentrations of ECG and KUR, as well as the combination? Could you explain in more detail how the determination of MIC combined is performed? Where did the authors get the numbers 32 and 4 in the first case (1199B strain)?
Table 2. Why are the results only found in 20 strains? and the others?
We have revised the results.
L.134. Delete "against and" from "with gentamicin against and reduced the"
L.156. delete "against three tested strains" or make sense to the sentence since it is repetitive.
L.157-158. The sentence is not understood. Please, it is important to rewrite it.
L.159. It is repetitive "with vancomycin (0.5 g/mL)" since it was mentioned before.
L.168. against what? complete the sentence or make sense.
L.182. Delete "and"
We have revised these sentences accordingly.
L.192 (Figure 3). Positive or negative? if positive, cytokine concentrations would be elevated in this group. Please check it out.
We have revised figures 3a and 3b to % inhibition of cytokine production from SEB and PGN stimulated PBMC, which would be better understood and prevent confusion to readers. For inhibitory effects on cytokine production from PGN or SEB-stimulated PBMC, dexamethasone is a positive control.
L.216. Why were antibiotics such as gentamicin or fusidic acid not tried in this animal model?
The MIC of gentamicin and fusidic acid for the corresponding resistant strains were relatively high when compared with vancomycin (0.5 mg/ml) for the tested strains. Therefore, we only chose vancomycin for our animal study. We may test gentamicin or fusidic acid in our future study.
L.241-242. This sentence is repetitive.
L.250-251. This sentence is repetitive.
We have revised these sentences section accordingly.
Discussion
The discussion is very deficient considering the results presented. Therefore, it has to be completely restructured considering the results obtained and having a much more scientific basis.
We found that the combination of ECG and kuraridin could enhance antibiotics activities in vitro but no significant enhancement with vancomycin was observed in in vivo studies. Therefore, we mainly discuss the main factor (bioavailability) which might affect the performance of ECG and kuraridin in animal testing. We further suggest some known and possible inhibitory mechanisms of ECG and kuraridin. For the revised discussion, we compared our results with some of the similar data from the literature (Triple combination of antimicrobials) and some further works that we are in progress.
L.262. Delete "additively".
L.265. Change "alone" to "separately"
L.271. low bioavailability of ECG and Kuraridin
L.274. Change "In this study" to "In that study"
We have revised these sentences accordingly.
Materials and methods
L.308-309. Why is it mentioned that Kanamycin, erythromycin and ciprofloxacin were tested as antibiotics and there are no results in the article? Who provided the antibiotics for this study? Include the reference with catalog number.
We have added the results of Kanamycin, erythromycin and ciprofloxacin in Table 2. The antibiotics used in the study are standard chemicals that were purchased from Sigma Chemical Company (St. Louis, MO, USA).
L.313. Change "The MIC of kuraridin is similar " to "The MIC of kuraridin was similar "
We have revised the sentence.
L.314. Include the reference.
We have added the reference in the text.
L.318. Please, be more explicit in the number of bacterial strains that were used since those found in the paragraph add up to 32 strains.
We have only included 29 strains in the study. W231, W233 and W238 are gentamicin-resistant strains, which were from our selected HA MRSA panel and appeared in two MIC tests.
L.352. Where did the authors get this criterion (FICI<0.5)? Please reference?
FICI<0.5 is defined as synergy. We have deleted the FICI<0.5.
L.360. Change "normal saline" to "saline".
We have revised the sentence.
L.360. Include the meaning of MH, please.
We have added Müller-Hinton (MH).
L.364. Why at 35 ºC if the optimum temperature for S. aureus is 37 ºC?
We have revised the temperature to 37 ºC.The optimal incubation temperature for S. aureus is 37 ºC. The use of 35+2 is suggested by the performance standards of clinical and laboratory standards institute.
L.374-376. It is important that the methodology be described in more detail, for example, the treatments (experimental groups, how many and which are they?) and the concentrations or doses at which the treatments were tested.
L.387. The plates were maintained in an incubator at 37 C and 5% CO2.
We have revised the PBMC methodology.
L.393. cell viability of at least 85% considered the compounds non-toxic. Why 85% and not 90% or 80%? What did the authors base their definition of 85% on?
We deleted the corresponding sentence. As kuraridin is not soluble at high concentrations (>64 mg/mL) and we could not obtain an IC50 value, 85% is just a suggested value for cytotoxicity. We have used growth inhibition in the revised results.
L.399-402. How many mice were used in the experiment? How many experimental groups did the experiment consist of?
L.408-411. Was it only 3 experimental groups? And the combinations to evaluate synergism? What were the combinations?
125 mice were included in the study. Five mice were used as normal control without infection and treatment. One hundred and twenty mice were equally divided into three groups (n=40 for each group) and infected with ATCC25923, ST30 or ST239. Each group of mice was subdivided into four treatment options (n=10 for each subgroup): No treatment, vancomycin (60 mg/kg) only, ECG (120 mg/kg) and kuraridin (120 mg/kg) and kuraridin (120 mg/kg), vancomycin (60 mg/kg) and ECG (120 mg/kg). Treatment and control regimens were initiated 2 h post-inoculation for every 12 hours two times daily for 2 days.
L.402-404. It is important that the way in which the treatments were prepared is described. For example, how was the Kuraridin solution prepared? Was only ethanol used? What are the concentrations of the solutions? Why was only vancomycin tested as an antibiotic (chemical entity)?
We have revised the methodology accordingly. Since ECG and kuraridin can lower the MIC of vancomycin to nanomolar concentration and the growth inhibition was bactericidal, we only chose vancomycin for animal study.
L.422. Change "on" to "at".
Changed.
L.422-423. Could the authors explain why the waiting times of 96h and 24h before carrying out the infection?
We have already explained this issue in the methodology. The ATCC25923 strain did not effectively infect in BALB/c mice; even the inoculations were at 1x109 CFU. For infecting ATCC25923 in BALB/c mice, neutropenia was induced in the mice prior to the infection using cyclophosphamide (intraperitoneally injection with 150 and 100 mg/kg cyclophosphamide at 96 h and 24 h before infection, respectively). After two doses of cyclophosphamide at 24 and 96 hours, the number of neutrophils will be significantly depleted after 96 hours and pneumonia can be induced by ATCC25923 infection.
Statistical analysis
L.447. Histopathological analyzes cannot be analyzed with parametric statistics since the variables are qualitative (categorical). Please carry out the corresponding analysis (NON-parametric statistics).
For Histopathological analyzes, we have revised the statistical analyses to the nonparametric Mann–Whitney test.
Conclusions
L.455-458. conclusion is not entirely true since in reality the combination of vancomycin with ECG and kuraridin did not present significant (synergistic) effects. Contradicts what is mentioned in L.216
We have revised our conclusion by deleting the second sentence in the paragraph to avoid contradiction.
Reviewer 2 Report
Manuscript Number antibiotics-2126365
"Enhancing antibiotics efficacy by combination of kuraridin and epicatechin gallate with antimicrobials against methicillin-resistant Staphylococcus aureus"
The paper provides a simple and novel method of using potential herbal combinations as adjuvants with conventional antibiotics to improve the efficacy of antibiotics. The study is well designed and the paper is very well written and has clinical applications. There are few minor revisions.
Please find below my comments:
- Figure 3a and 3b, see if you can represent in a better understandable way. Fig 3c, check if y axis legend is correct?
- Represent reduction of lesions in figure 5 using arrows, and explain the histopathology images in more detail. Figure 5b looks blur.
- Discussion is more or less like a summary of results. Discussion is very brief and lacks comparison of obtained results with the literature. So, authors are asked to discuss each of the results with relevant references.
- Can the authors explain what is that MIC combine in Table 1 and 2?? Why is MIC combine named same as MIC as ECG and Kur only. Can it be named differently.
- In Table 2, MIC alone, ECG as 2 values of MIC alone?? Can u explain that in detail.
- Since this is a journal of antibiotics, authors are requested to explain time kill study in detail (methodology section) instead of just mentioning previously described.
- Mention the units for MIC in all the Tables or in the Table figure legends.
- Can the authors explain why XTT assay was used wherein MTT assay is widely used to do the cytotoxicity studies??
- Mention what is the future scope of the study or studies that need to be carried out in future under the conclusion section, which provides opportunity for the researchers to continue with the study for further clinical implications.
- Recheck for minor grammatical errors and English language at few places.
Author Response
Reviewer 2
We appreciate the reviewer comments and constructive feedback related to the manuscript. We revised the manuscript according to these comments and provide detailed answers to the comments below. All changes in the manuscript can be tracked via the track change function.
- Figure 3a and 3b, see if you can represent in a better understandable way. Fig 3c, check if y axis legend is correct?
We have revised figure 3a and 3b to % inhibition of cytokine production from SEB and PGN stimulated PBMC, which would be better understandable to readers. The y axis of figure 3c is % of control (The control is OD of drug free control).
- Represent reduction of lesions in figure 5 using arrows, and explain the histopathology images in more detail. Figure 5b looks blur.
It is difficult to represent reduction of lesions. For the treatment groups, less lesion areas were observed when compared with no treatment groups. The features of lesion include: extension of inflammatory cell infiltration and red blood cells from bronchioles and perivascular areas into the surrounding lung parenchyma was apparent with several focal areas of heavy consolidation becoming larger and more diffuse.
We have added more detail descriptions for the lung histopathology images in the results section and we have revised Figure 5b.
- Discussion is more or less like a summary of results. Discussion is very brief and lacks comparison of obtained results with the literature. So, authors are asked to discuss each of the results with relevant references.
We found that the combination of ECG and kuraridin could enhance antibiotics activities in vitro but no significant enhancement with vancomycin was observed in in vivo studies. Therefore, we mainly discuss the main factor (bioavailability) which might affect the performance of ECG and kuraridin in animal testing. We further suggest some known and possible inhibitory mechanisms of ECG and kuraridin. For the revised discussion, we compared our results with some of the similar data from the literature (Triple combination of antimicrobials) and some further works that we are in progress.
- Can the authors explain what is that MIC combine in Table 1 and 2?? Why is MIC combine named same as MIC as ECG and Kur only. Can it be named differently.
MIC alone is the MIC value of that drug when used alone against a tested strain, MIC combine the MIC value of that drug which was combined with other drug(s) against a tested strain. We have added explanation of the MIC values in results and methods section.
- In Table 2, MIC alone, ECG as 2 values of MIC alone?? Can u explain that in detail.
We have revised Table 2 to Table 3-5, the values in the first row were ECG and kuraridin combined with antibiotics, and the values of the second row were only ECG and kuraridin in combination
- Since this is a journal of antibiotics, authors are requested to explain time kill study in detail (methodology section) instead of just mentioning previously described.
We have revised the protocols of time –kill study in more detailed.
- Mention the units for MIC in all the Tables or in the Table figure legends.
We have added the units for MIC in the tables.
- Can the authors explain why XTT assay was used wherein MTT assay is widely used to do the cytotoxicity studies??
Since the cells in PBMC are not all adherence cells (e.g. B and T cells) and some cells will be lost during MTT assay (remove the cell supernatant and the MTT solution during assay) and affect the results. We therefore used XTT in our study, which could prevent non-adherent cell loss.
- Mention what is the future scope of the study or studies that need to be carried out in future under the conclusion section, which provides opportunity for the researchers to continue with the study for further clinical implications.
We have added our future studies in the discussion
- Recheck for minor grammatical errors and English language at few places.
We have revised the manuscript carefully.
Reviewer 3 Report
Chan et al.'s study investigated the antimicrobial efficacy of combining two natural compounds (kuraridin and 2 epicatechin gallate) with antibiotics against MRSA strains. The study found that this combination could provide the best synergy with antibiotics against tested bacteria. Further, this natural compound has anti-inflammatory properties. This study is fascinating, relevant, and well writing due to the rising threat of antimicrobial resistance. Triple combinations of antibiotics could be an excellent option to overcome the challenge of AMR. However, minor issues must be corrected before considering this manuscript for publication.
1. The English throughout the manuscript must be improved
2. There are some typo errors. Please correct it
3. Please put the table title in numerical order as table 1,2…, and report each table separately.
4. In the time-kill assay, did the author only test each isolated one time?
5. Please remove the double parentheses. E.g., lines 85, 87, 91 … etc.
6. Line 99: (>995) please add space between the number and symbol.
7. Line 430: For pneumonia assessment, please cite the reference for this method
8. The antibacterial activities of kuraridin may involve inhibiting some key enzymes in MRSA for survival. Please explains more and add some recent reference
Author Response
We appreciate the reviewer's comments and constructive feedback related to the manuscript. We revised the manuscript according to these comments and provide detailed answers to the comments below. All changes in the manuscript can be tracked via the track change function.
- The English throughout the manuscript must be improved
We have revised our manuscript carefully.
- There are some typo errors. Please correct it
We have corrected the typo errors.
- Please put the table title in numerical order as table 1,2…, and report each table separately.
We have revised the tables and the table title in numerical order.
- In the time-kill assay, did the author only test each isolated one time?
The results for the time-kill assays were collected from 3 independent experiments.
- Please remove the double parentheses. E.g., lines 85, 87, 91 … etc.
We have removed the double parentheses.
- Line 99: (>995) please add space between the number and symbol.
We have added s space between the number and the symbol.
- Line 430: For pneumonia assessment, please cite the reference for this method
We have added the following reference for pneumonia assessment:
Labandeira-Rey, M.; Couzon, F.; Boisset, S.; Brown, E. L.; Bes, M.; Benito, Y.; Barbu, E. M.; Vazquez, V.; Hook, M.; Etienne, J.; Vandenesch, F.; Bowden, M. G., Staphylococcus aureus Panton-Valentine leukocidin causes necrotizing pneumonia. Science 2007, 315, (5815), 1130-3.
- The antibacterial activities of kuraridin may involve inhibiting some key enzymes in MRSA for survival. Please explains more and add some recent reference
We have added one example with reference to the discussion. One of the target enzymes is sortase A. In S. aureus, more than 20 distinct surface virulence factors are anchored to the cell wall by the extracellular sortase A (SrtA) enzyme.
Round 2
Reviewer 1 Report
Please review the attached file. There are some minimal suggestions that can improve the quality of the manuscript.

Author Response
We appreciate the reviewer's comments related to our revised manuscript. We revised the manuscript and all changes in the manuscript can be tracked via the track change function.